# Learning from conect4children: A Collaborative Approach towards Standardisation of Disease-Specific Paediatric Research Data

Anando Sen [1], Victoria Hedley [1], Eva Degraeuwe [2], Steven Hirschfeld [3], Ronald Cornet [4], Ramona Walls [5], John Owen [6], Peter N. Robinson [7], Edward G. Neilan [8], Thomas Liener [9], Giovanni Nisato [9], Neena Modi [10], Simon Woodworth [11], Avril Palmeri [1], Ricarda Gaentzsch [12], Melissa Walsh [11], Teresa Berkery [11], Joanne Lee [1], Laura Persijn [2,13], Kasey Baker [8], Kristina An Haack [14], Sonia Segovia Simon [1], Julius O. B. Jacobsen [15], Giorgio Reggiardo [16], Melissa A. Kirwin [17], Jessie Trueman [1], Claudia Pansieri [16], Donato Bonifazi [16], Sinéad Nally [18], Fedele Bonifazi [19], Rebecca Leary [1,*] and Volker Straub [1]

1 John Walton Muscular Dystrophy Research Centre, Newcastle University, Newcastle upon Tyne NE1 3BZ, UK
2 Ghent University, 9000 Ghent, Belgium
3 Uniformed Services University of the Health Sciences, Bethesda, MD 20814, USA
4 Amsterdam University Medical Centers, 1105 AZ Amsterdam, The Netherlands
5 Critical Path Institute, Tucsan, AZ 85718, USA
6 Clinical Data Interchange Standards Consortium (CDISC) Europe Foundation, Saint-Gilles, 1060 Brussels, Belgium
7 The Jackson Laboratory for Genomic Medicine, Farmington, CT 06032, USA
8 National Organization for Rare Disorders, Quincy, MA 02169, USA
9 Pistoia Alliance, Wakefield, MA 01880, USA
10 Imperial College London, London SW7 2AZ, UK
11 INFANT Research Centre, University College Cork, Cork T12 YE02, Ireland
12 IQVIA Italy S.r.l., 20124 Milan, Italy
13 Belgian Pediatric Clinical Research Network, Ghent University Hospital, 9000 Ghent, Belgium
14 Sanofi Genzyme, 75017 Paris, France
15 Queen Mary University of London, London E1 4NS, UK
16 Consorzio per Valutazioni Biologiche e Farmacologiche, 70122 Bari, Italy
17 Clinical Data Interchange Standards Consortium (CDISC), Austin, TX 78701, USA
18 Novartis Pharmaceuticals, Dublin D04 NN12, Ireland
19 Fondazione per la Ricerca Farmacologica Gianni Benzi Onlus, 70124 Bari, Italy
* Correspondence: becca.leary@newcastle.ac.uk

**Abstract:** The conect4children (c4c) initiative was established to facilitate the development of new drugs and other therapies for paediatric patients. It is widely recognised that there are not enough medicines tested for all relevant ages of the paediatric population. To overcome this, it is imperative that clinical data from different sources are interoperable and can be pooled for larger post hoc studies. c4c has collaborated with the Clinical Data Interchange Standards Consortium (CDISC) to develop cross-cutting data resources that build on existing CDISC standards in an effort to standardise paediatric data. The natural next step was an extension to disease-specific data items. c4c brought together several existing initiatives and resources relevant to disease-specific data and analysed their use for standardising disease-specific data in clinical trials. Several case studies that combined disease-specific data from multiple trials have demonstrated the need for disease-specific data standardisation. We identified three relevant initiatives. These include European Reference Networks, European Joint Programme on Rare Diseases, and Pistoia Alliance. Other resources reviewed were National Cancer Institute Enterprise Vocabulary Services, CDISC standards, pharmaceutical company-specific data dictionaries, Human Phenotype Ontology, Phenopackets, Unified Registry for Inherited Metabolic Disorders, Orphacodes, Rare Disease Cures Accelerator-Data and Analytics Platform (RDCA-DAP), and Observational Medical Outcomes Partnership. The collaborative partners associated with these resources were also reviewed briefly. A plan of action focussed on collaboration was generated for standardising disease-specific paediatric clinical trial data. A paediatric data standards multistakeholder and multi-project user group was established to guide

the remaining actions—FAIRification of metadata, a Phenopackets pilot with RDCA-DAP, applying Orphacodes to case report forms of clinical trials, introducing CDISC standards into European Reference Networks, testing of the CDISC Pediatric User Guide using data from the mentioned resources and organisation of further workshops and educational materials.

**Keywords:** conect4children; paediatric clinical trials; data standards; FAIR principles; interoperability; CDISC

## 1. Introduction

The conect4children (c4c) project (funded by the Innovative Medicines Initiative 2 Joint Undertaking, a public–private partnership between the European Union and the European pharmaceutical industry) was established to overcome the many barriers to the planning and delivery of effective and efficient paediatric clinical trials [1]. Such barriers include regulatory timelines, age-appropriate formulations and dosages, different ages of consent in different countries, scarce and specific populations, site-specific considerations, and ethical issues [2–6]. The c4c project stems from the fundamental recognition that there is not enough medicine developed for and tested in all relevant ages of the paediatric population and that safety and efficacy data are correspondingly scarce [7–9]. There is a general and growing awareness of the need to use and reuse data optimally from vulnerable populations. One route to better designed, faster, and more streamlined clinical trials in paediatric diseases is an enhanced ability for data collected in disparate systems to be able to 'speak' with other data. This form of interoperability will enable data collected across different platforms to be pooled together for larger post-hoc analyses, enhance statistical power, analysis of rare adverse events, subgroup analyses for under-represented subgroups, pragmatic clinical trials, and possible comparator arms [10,11]. Such an approach began in the 1950s in paediatric oncology and extended to some other types of subspecialities, primarily supported by subspeciality networks [12]. The consequence has been the growth of multiple independent parochial data ecosystems.

Pooling of data is particularly relevant in paediatrics as well as in any other small populations, including the rare disease field. There are important overlaps between these communities. Although not all paediatric conditions are rare, most rare diseases occur in the paediatric population, and, therefore, it is important that c4c, as an overarching optimalisation of paediatric clinical trials organisation, is able to recognise, anticipate, and manage the multiple complexities resulting from delivering trials in rare diseases [13–16].

c4c has prioritised developing resources to enable the pooling of data. The c4c project includes 35 academic and 10 industry partners. At the beginning of the project, the data work package leads devised a plan to tackle well-known data issues with a focus on the standardisation and harmonisation of paediatric clinical trial data. The plan included the following: (1) the development of a cross-cutting paediatric data dictionary (CCPDD); (2) the development of a paediatric user guide (PUG) in collaboration with the Clinical Data Interchange Standards Consortium (CDISC); (3) a pathway for extension of the CCPDD and PUG to disease-specific data items; (4) further extensions to real-world data (RWD); and (5) work towards Findable, Accessible, Interoperable, and Reusable (FAIR) data in all the previous tasks.

The CCPDD was developed over 18 months through a consensus-based process. It includes 25 cross-cutting paediatric data items with references to CDISC standards wherever available. As several countries require CDISC standards for regulatory submissions, the CCPDD was made available for use immediately upon release in late 2019. It has since been used in three clinical trials [17]. Future editions of the CCPDD (currently under development) will include additional terms, references to additional data standards and could possibly be based on a clinical modelling tool for better visualisation [18]. The CCPDD served as one of the inputs for the CDISC PUG. The focus of the PUG remained on

cross-cutting paediatric data, though it contained more items than the CCPDD. As with the CCPDD, it was developed through a scoping and consensus-based process. The PUG is owned by CDISC and contains examples and guidance on implementing CDISC standards for a variety of uses, including global regulatory submissions. As of February 2023, the PUG is publicly and freely available on the CDISC website [19].

Disease-specific data standardisation is a natural extension of cross-cutting standardisation. High off-label drug usage in children continues to persist, in part due to the challenges and delays of numerous clinical trials in disease-specific indications [20]. The utility of disease-specific standardisation is exemplified by previous studies, which pooled data from a small number of studies to draw important clinical, safety, and patient management conclusions. Randeree et al. considered pooled data from four trials to analyse the safety and efficacy of Eteplirsen for Duchenne Muscular Dystrophy patients [21]. This study demonstrated no statistically significant differences, but it recommended more clinical trials. Eventually, the drug was refused marketing authorisation by the European Medical Association (EMA), despite having been approved by the Food and Drug Association (FDA) previously [22]. Bird et al. compared the dose-dependent incidence of acute kidney injury in paediatric thalassaemia patients treated with Deferasirox in ten clinical studies [23]. Results showed that the drug could cause acute kidney injury in a dose-dependent manner and called for regular monitoring of renal function and serum ferritin in treated patients. Nast et al. examined the trial dropout timeframes due to lack of efficacy in psoriasis patients treated with Etanercept using data from 10 trials [24]. The mean time to drop out was about 80 days, showing this as a critical timeframe within which psoriasis patients expect effective treatment.

A strategy needs to be developed on how to approach data harmonisation in a disease-specific setting, including (1) clinical study data collected in different clinical trials by the same sponsor; (2) clinical study data from different studies led by different sponsors; and (3) clinical study data linked with real-world data (RWD) such as electronic health records (EHR) or registry data. In this paper, we review major initiatives and resources that can aid disease-specific standardisation. We then present an implementation plan for disease-specific standardisation based on these resources. While the extension to RWD is not strictly a part of this paper (c4c has a separate working group addressing RWD), a lot of the resources presented here may be applicable to RWD.

## 2. Methods

c4c brought together international stakeholders from academia, industry, data standards consortia, data repositories, and large paediatric health networks to evaluate the feasibility of disease-specific data standardisation. Due to the sheer volume of disease-specific resources, the focus remained on the use of existing resources that could aid in eventually achieving standardisation. These include resources that enable pooling of data (e.g., registries), promote interoperable data through common data standards, increase the FAIRness of data, develop data dictionaries, and maintain repositories for clinical data. The initiatives and resources are discussed below. Additional resources that collaborate with these resources have also been briefly discussed. The level of detail about each resource is not representative of its importance in the field, and the list is not meant to be exhaustive.

The resources can broadly be classified into three categories—(1) large consortia or initiatives that are promoting standardisation, interoperability, and reusability of health data and developing tools for the purpose; (2) data sources such as repositories or registries; and (3) organisations developing data standards and/or maintaining data dictionaries. These categorisations are shown in Figure 1. While there may be overlaps among these categories, the resource is described under the most relevant category.

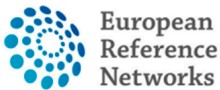
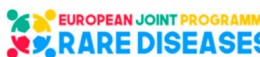
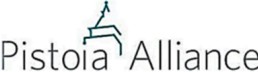
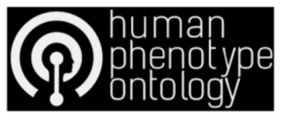
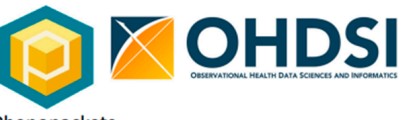
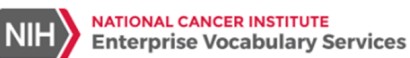
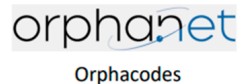
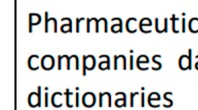
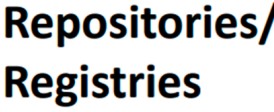
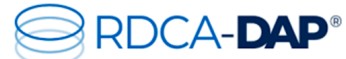
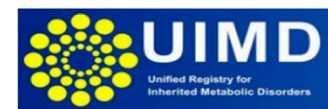
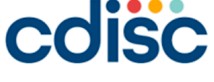

**Initiatives**

**Data Standards and Dictionaries**

**Repositories/ Registries**

**Figure 1.** The categorisations for the 13 initiatives and resources brought together by c4c for disease-specific standardisation.

*2.1. Large Initiatives*

2.1.1. European Reference Network (ERN)

An ERN facilitates virtual connectivity between transnational healthcare providers to maximise the reach of a provider's expertise. Through an ERN, a clinician can access expert knowledge about a patient's disease, no matter where the knowledge was generated. This is particularly important in rare diseases where expertise may be limited.

The 24 ERNs aim to tackle complex diseases by establishing pseudo-anonymised registries with patient-level data. Each ERN tackles a different therapeutic area (e.g., neurological disease, paediatric cancers) [25]. ERNs collaborate with other European projects, including the European Joint Program on Rare Disease in increasing the FAIRness of data. Some ERNs can include longitudinal data and disease-specific sub-registries. ERNs use the concept of 'common data element' (CDE) in their data dictionaries [26]. A CDE refers to patients' personal data, diagnosis, history, care pathway, disability, and informed consent. Sixteen CDEs are mandatory for every rare disease registry in Europe. ERN data dictionaries vary greatly in size, containing between sixteen and over a thousand data elements.

ERKNet (European Reference Network on kidney diseases) is an ERN that has built the ERKReg (European Rare Kidney Disease Registry) [27]. The registry contains sub-registries for specific kidney diseases and follow-up data is collected on a yearly basis. ERKReg uses ontologies to describe its data elements—Orphacode for clinical diagnoses, Human Genome Variation (HGV) for genetic diagnoses, Anatomical Therapeutic Chemical (ATC) for medications, and Human Phenotype Ontology (HPO) for phenotypes.

2.1.2. European Joint Programme on Rare Disease (EJP RD)

The EJP RD (funded by the European Union's Horizon 2020 research and innovation programme under grant agreement N°825575) was set up to tackle the fragmented nature of rare disease research across Europe. It is made up of 130 institutions based in 35 countries and includes all 24 ERNs [28]. One of the pillars of EJP RD has been the development of a virtual platform that would be a single access point for registries, repositories, libraries, biobanks, analysis platforms, and supporting materials for rare diseases. While the data will remain at the source, both humans and machines will be able to query the rare disease resources through the platform [29].

Another resource produced within the EJP RD is the Innovation Management Toolbox (IMT). The IMT is a reference library of resources in rare disease translational medicine. It

is freely available and accessible via the EJP RD website [30]. Other pillars of EJP RD focus upon aggregating funding opportunities, developing training materials, and providing clinical trial support [29].

### 2.1.3. Pistoia Alliance

The Pistoia Alliance is a non-profit organisation, founded in 2007 by four large pharmaceutical companies, with the mission to facilitate pre-competitive collaboration in the life science industry by conducting joint projects that share risks and rewards. Its members also comprise technology and solution providers, academic institutions, publishers, and patient organisations.

The Pistoia Alliance manages diverse communities of interest, organises conferences and events, facilitates best practice definition and sharing, and enables the co-creation of shared resources through collaborative projects. The Pistoia Alliance FAIR implementation project team developed the FAIR Toolkit, which highlights methods as well as use cases of FAIR implementation in industry [31,32]. The project team also collaborated with another EU project, FAIRplus, to align and link the FAIR Toolkit and FAIRplus' FAIR cookbook [33]. Another important resource developed by the project team is the FAIR4Clin guide, which focuses on clinical trial data and real-world data [34] and aims to help organisations start addressing the FAIR guiding principles in the clinical setting.

A summary of these three resources is presented in Table 1.

**Table 1.** A summary of the resources of large initiatives promoting standardisation, interoperability and reusability of health data and developing tools for the purpose.

| Initiative | Resource | Description |
|---|---|---|
| ERNs | Registries | 24 pseudo-anonymised registries with patient-level data, each tackling a different therapeutic area |
| | Data dictionaries including Common Data Elements (CDE) | Can contain thousands of elements related to the registry including 16 mandatory CDE |
| EJPRD | Single-point access Virtual Platform | Single access point for registries, repositories, libraries, biobanks, and analysis platforms related to rare disease |
| | Innovation Management Toolbox | Library of self-help resources in rare disease translational medicine |
| Pistoia Alliance | FAIR Toolkit | Highlights methods and use cases of FAIR implementation in industry |
| | FAIR4clin guide | FAIR guiding principles for clinical trial and real-world data |

### 2.1.4. Other Initiatives of Relevance to Paediatrics

The European Clinical Research Infrastructure Network (ECRIN) facilitates clinical research across European nations through advice and services with a long-term vision of generating evidence to optimise medical practice. It has 13 member countries and 120 clinical trial units. Services provided by ECRIN relate to the preparation, setup, and management of clinical trials [35,36]. Another initiative, Screen4Care, calls for a novel approach for diagnosing rare diseases and includes 36 partners in 13 countries [37]. The European Paediatric Translational Research Infrastructure (EPTRI) provides preclinical and translational research services in paediatrics to be translated into clinical research and paediatric use of medicines [38].

Certain initiatives cover specific diseases. While they may not directly address standardisation issues, their knowledge base is crucial for addressing disease-specific standardisation. The Paediatric European Network for Treatment of AIDS (PENTA) was established as a collaboration between paediatric HIV centres but has expanded to other infectious diseases [39]. The European Society for Paediatric Oncology (SIOPE) is a European organisation representing paediatric cancer experts. It aims to increase the cure rate and the quality

of a cure for children with cancer over the next decade [40]. The Paediatric Rheumatology International Trials Organisation (PRINTO) is an international network with the goal of facilitating, conducting, and analysing global clinical trials and outcome standardisation for paediatric rheumatic disease patients [41]. The European Cystic Fibrosis Society–Clinical Trials Network (ECFS-CTN) provides access to 57 CF centres in 17 countries. Besides providing a dedicated network, ECFS-CTN aims to standardise protocol reviews, research procedures, and outcome measures [42]. TREAT-NMD (Translational Research in Europe for the Assessment and Treatment for Neuromuscular Disorders) is a global network that provides the infrastructure for novel therapies to reach NMD patients quickly [43]. The International Hemoglobinopathy Research Network (INHERENT) aims to study the role of genetic modifiers in hemoglobinopathies through a large, multi-ethnic genome-wide association study. The target sample size of INHERENT is at least 30,000 individuals with hemoglobinopathies [44].

## 2.2. Data Repositories and Registries

### 2.2.1. Rare Disease Cures Accelerator—Data Analytics Platform (RDCA-DAP)

RDCA-DAP contains rare disease patient-level data from multiple sources. RDCA-DAP is housed at the Critical Pathology Institute (C-Path). C-Path has partnered with the National Organization for Rare Disorders (NORD) to leverage its IAMRARE registry platform. NORD also brings significant experience in identifying potential data-contributing organisations and establishing contact with them. C-Path has extensive experience in data curation, data pooling, and data analysis, which allows them to deal with common data quality issues such as missing values, standardisation and harmonisation issues, and problems with data dictionaries [45]. RDCA-DAP is standards-agnostic but currently uses CDISC and Observational Medical Outcome Partnership (OMOP) standards as well as Open Biomedical and Biological Ontologies (OBO) Foundry and other ontologies to integrate data within and across disease areas. C-Path is collaborating with EJP RD to align with the ontologies developed by the ERNs.

When a user requests data, the data access (if approved) is dependent upon the degree of data sharing agreed to by the data contributors. In most cases, the user operates on the data within workspaces in a trusted research environment where no data can leave the environment without a proper approval process [46,47], but users may bring in their own private data to combine with RDCA-DAP data.

### 2.2.2. Unified European Registry for Inherited Metabolic Disorders (U-IMD)

The U-IMD is an initiative to bring together all existing IMD registries. The initiative is in collaboration with the European Reference Network for Hereditary Metabolic Disorders (MetabERN). U-IMD is developing a standard consisting of minimal core data sets in MetabERN and the aforementioned ERKNet. The long-term goal of U-IMD is to improve health outcomes among patients of all ages who are affected by rare IMDs. The first registry brought under U-IMD was the registry of the International Working Group on Neurotransmitter-Related Disorders (iNTD) [48].

### 2.2.3. Other Repositories

Repositories with clinical trials and real-world data have become more widely available for public usage in the past decade. It is important to note, however, that in most cases, a data access request is required, and the process to finally obtain the data can take several months. Depending on the data sharing agreement, data access may be restricted to secure research environments.

The UK National Neonatal Research Database contains a standard data extract from the Electronic Patient Records of all admissions to National Health Service neonatal units from 2007 to the present. Data are quality-assured and curated prior to entry into the database and are mapped to the OMOP Common Data Model (see below). Access processes are through the Health Data Research UK Alliance Gateway [49]. This is a rare case of a strictly

paediatric repository. Hence, repositories that primarily contain adult patient data but have provisions for paediatric data are of specific interest. The review and analysis of such repositories is being carried out in a parallel c4c study. While going over every analysed repository is beyond the scope of this paper, a few examples are discussed below.

The Yale University Open Data Access (YODA) consists of data from 445 clinical trials that have led to 97 publications (from 84 different projects). The Vivli global data-sharing and analytics platform contains data from over 6600 trials consisting of over 3.5 million patients. Clinical Study Data Request (CSDR) is a platform for clinical trial sponsors to securely share patient-level data. As of May 2023, 3052 studies are available on the platform, with 772 data requests since 2013 (leading to 114 publications). While these data repositories are not specifically paediatric, they contain substantial paediatric data.

### 2.3. Data Standards and Dictionaries

2.3.1. Clinical Data Interchange Standards Consortium (CDISC) Therapeutic Area Standards

The CDISC standards are required for regulatory submissions in the USA and Japan and are recommended in China [50–52]. CDISC standards are undergoing pilot studies in Europe and could be adopted once the results of the studies are available [53]. The CDISC Therapeutic Area (TA) Standards extend the CDISC foundational standards to represent data that pertains to specific disease areas. TA standards include disease-specific metadata, examples, and guidance on implementing CDISC standards for a variety of uses, including global regulatory submission. To date, CDISC has developed and published 49 Therapeutic Area User Guides (TAUGs). CDISC has developed the previously mentioned PUG in collaboration with c4c and is working on the rare disease TAUG in collaboration with NORD.

While rare diseases encompass multiple TAs, the diseases share a lot of similarities. The rare disease TAUG will facilitate the adaptation of CDISC standards by rare disease researchers so that more rare disease data are collected in a standardised manner and can be readily submitted to the FDA and other regulatory agencies [54].

2.3.2. Observational Medical Outcomes Partnership (OMOP) Common Data Model (CDM)

The OMOP CDM enables the standardisation of observational data for efficient analyses. The OMOP CDM was developed by the Observational Health Data Science and Informatics (OHDSI) consortium. The OHDSI vocabularies allow the organisation of terminologies to be used in the CDM [55]. Standardised analytics tools are also available.

The utility of the OMOP CDM has been demonstrated by bringing together data from different sources into a common format [56,57]. Standards such as OMOP CDM and Health Level 7 (HL7) Fast Healthcare Interoperability Resources (FHIR) are important for bridging the gap between RWD and clinical trial data, as CDISC standards are not well understood in academic and healthcare settings [58]. For example, C-Path has used OMOP to standardise several of their clinical trial data sets.

2.3.3. National Cancer Institute (NCI) Enterprise Vocabulary Service (EVS)

EVS is responsible for developing and curating biomedical terminology that is needed by the NCI and its wider community. EVS works with many partners to develop, licence, and publish terminology, develop software tools, and support shared standards. EVS provides the foundational layer for NCI's data representation and translation within NCI's informatics infrastructure. EVS developed the NCI Thesaurus, which serves as its database for reference terminologies [59,60].

The NCI EVS is the repository of record for a multi-year project to harmonise paediatric subspeciality data, beginning with a catalogue of common adverse events as applied to paediatric populations developed by a multidisciplinary international team in collaboration with the Medical Dictionary for Regulatory Activities (MedDRA—see below). Subsequently,

the terms were integrated into MedDRA and are suitable and acceptable for regulatory submissions [61,62].

NCI EVS and CDISC have been in partnership for over 20 years [63]. While EVS is responsible for providing expertise in terminology, subject matter, and management, CDISC is responsible for data modelling, standards development processes, and volunteer reviews.

### 2.3.4. Human Phenotype Ontology (HPO) and the GA4GH Phenopacket Schema

The HPO is an ontology of medicine-related phenotypes, annotations, and operating algorithms [64]. The HPO can be used for translational research and applications in computational biology as it can 'compute over clinical phenotype'. Applications include deep phenotyping, which is the analysis of phenotypic abnormalities where the components of the phenotype are analysed [65]. The HPO has been used by multiple registries, software, and organisations for describing phenotypic abnormalities [66].

HPO can be easily combined with the Phenopacket Schema, which is a standard developed under the aegis of the Global Alliance for Genomics and Health (GA4GH). Many healthcare systems rely on manual or institution-specific entry of data, making it difficult to exchange health information electronically. The Phenopacket Schema can provide a common format for integrating an individual's phenotypic abnormalities with disease, measurements, interpretations, genotype, and other data or metadata [67]. This can be adapted for multiple patients, diseases, and institutions, enabling meaningful comparisons among a wide and diverse range of subjects.

For paediatric clinical trials, Phenopackets could play an important role in standardising case report forms (CRFs). Currently, CRFs can be highly heterogenous, with little to no scope for long-term interoperability. Implementations have been developed for Phenopackets to enable integration with real-world data, FHIR, and OMOP [68].

### 2.3.5. Orphacodes

Orphacodes are a nomenclature system for rare diseases maintained by Orphanet. The need for Orphacodes was necessitated by a severe lack of ICD-10 codes for rare diseases. As of 2010, only 355 of the 6000–8000 rare diseases had a unique ICD-10 code compared to the 6954 clinical entities listed by Orphanet. While this number grew to over 5400 for ICD-11, it will be several years before ICD-11 is widely adopted [69]. Orphanet maintains a multilingual nomenclature of rare diseases, which supports its other relational databases. Orphacodes are unique, non-reusable, and aligned with other standard terminologies, including ICD, SNOMED-CT, UMLS, etc. These cross-maps aid interoperability.

### 2.3.6. Industry Data Dictionaries

Data dictionaries developed by pharmaceutical companies for internal usage are proprietary and not available publicly. Their development and usage were briefly addressed when c4c industry partners were interviewed in the context of developing academia–industry collaborations [70]. All four of the interviewed companies use CDISC standards as far as possible but need to expand their own dictionaries, according to guidelines laid out in the CDISC implementation guides, whenever a data item cannot be represented in published CDISC standards. This is particularly true for paediatric data items, as CDISC standards are known to lack paediatric specificity [71,72]. In such cases, all companies had strict processes for adding data items that are not represented in published CDISC standards to their internal dictionaries. In fact, some companies officially submit requests to CDISC to add any data items that were identified as not available in CDISC.

### 2.3.7. Other Data Standards and Dictionaries Relevant to Paediatrics

Although not paediatric-specific and, in fact, generally deficient in paediatric terms, several computer-processable terminology collections are in global use, each with a specific purpose. While CDISC serves as a resource primarily for research data, other resources focus primarily on healthcare delivery, reimbursement, or epidemiology.

Health Level 7 (HL7) functions as a framework for electronic health information exchange and healthcare delivery management. Within HL7, the Fast Healthcare Interoperability Resources (FHIR) is a widely used EHR data standard. FHIR builds on HL7 standards, and the data can be represented in XML, JSON, and RDF formats. One of the major goals of the FHIR is to facilitate interoperability between modern and legacy healthcare systems. Another high-level initiative is Integrating the Healthcare Enterprise (IHE), which develops workflow-based interoperability specifications among its activities. IHE profiles provide exact definitions of implantation standards for specific clinical needs.

Additional resources can be categorised using a general workflow, beginning with clinical observations that lead to diagnoses that trigger services, procedures, and administrative management and concluding with data submitted for reimbursement. Two major observational standards are Logical Observation Identifiers Names and Codes (LOINC) and Digital Imaging and Communications in Medicine (DICOM). LOINC is a system for recording and transmitting laboratory test results and other types of observations. The terminology and code list were originally developed for laboratory observations and have since been expanded to other clinical domains, such as vital signs. DICOM is a standard for digital medical images such as computed tomography and nuclear medicine.

Observational data are organised and analysed to support diagnoses. The two major terminology systems used for diagnoses are the Systemized Nomenclature of Medicine Clinical Terms (SNOMED CT) and the International Classification of Disease (ICD). SNOMED CT is an organised collection of medical terms curated by an international coalition that provides codes, synonyms, definitions, etc. ICD is a classification system endorsed by the World Health Organization (WHO) that describes diseases and procedures. It is highly nuanced and can include symptoms, additional abnormalities, social circumstances, and external causes. ICD is used in some regions to code individual patient care and for reimbursement.

For billing and reimbursement, the terminologies vary by region. For example, in the United States, the American Medical Association supports the Current Procedural Terminology (CPT) for coding services, procedures, administrative management, and developing guidelines for medical care review. CPT codes, combined with ICD codes, are the primary means to record patient care activity for both federal government and third-party payer reporting and reimbursement.

For regulatory submissions, the standard is the Medical Dictionary for Regulatory Activities (MedDRA). MedDRA is a medical terminology resource developed by the International Council for Harmonization of Technical Requirements for Pharmaceuticals for Human Use (ICH) for data sharing of regulatory information.

In addition to the NCI EVS, the Unified Medical Language System (UMLS), curated by the US National Library of Medicine, integrates and distributes key terminology, classification and coding standards, and associated resources to promote effective and interoperable biomedical information systems and services, including electronic health records. The UMLS is a set of files and software that brings together many health and biomedical vocabularies and standards to enable interoperability between computer systems.

As noted previously, the commonly used data and terminology resources were developed and remain grounded and oriented to healthcare provision and activities for adult populations. Figure 2 provides a schematic of the relationship among these entities with exemplars but not a comprehensive listing of use cases. In addition, most of the major resources have relationships and mapping to each other either directly or through intermediaries such as FIHR, NCI EVS, and the UMLS.

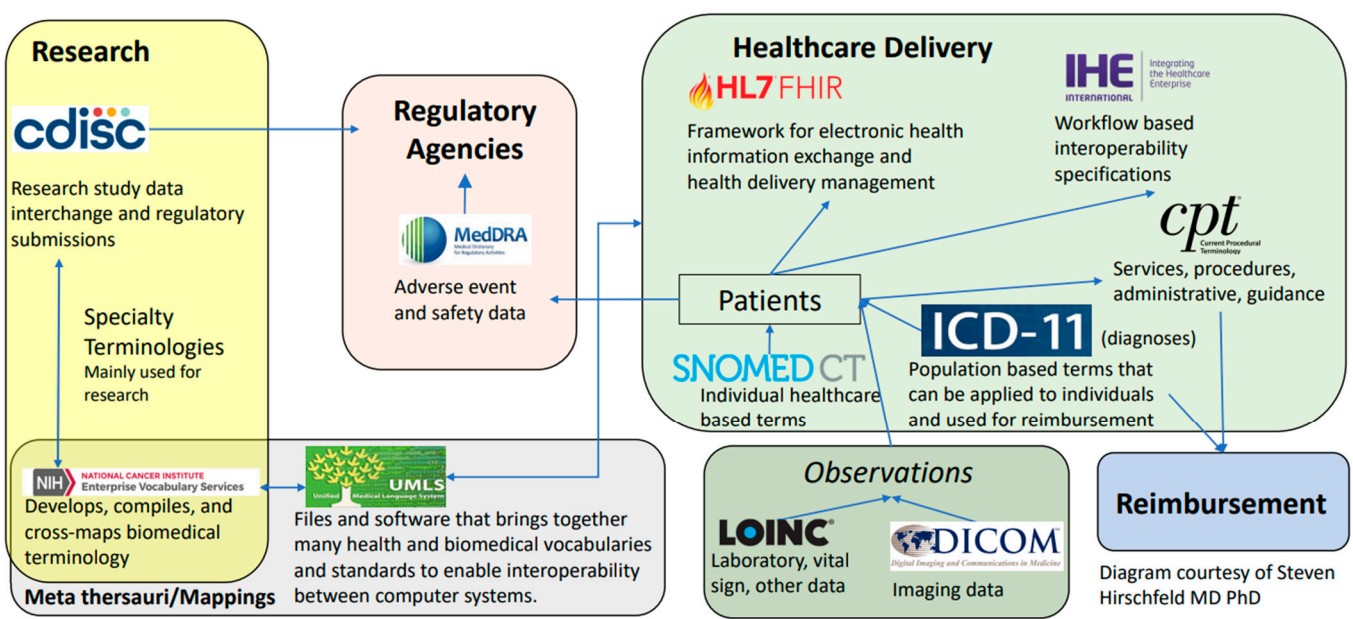

**Figure 2.** Exemplar relationships between select terminologies, data standards, and resources. The spheres of activity are designated by coloured boxes where yellow is research, grey is meta thesauri and mappings, light orange is regulatory activities, green is healthcare delivery (slightly darker for a sub-box of observations), and blue is reimbursement. The mappings are simplified and do not indicate the many levels of interactivity and potential interoperability.

## 3. Results

A major theme that emerged from investigating these resources was the willingness and need to maximise collaborative efforts (Figure 3). As mentioned in the Introduction, paediatric clinical research is already challenging due to scarce and specific populations. The challenge becomes greater when the data pool is further reduced by disease-specificity. To offset this, collaboration between all concerned parties is essential, and many collaborations are already ongoing (Table 2).

**Table 2.** Existing collaborations between various initiatives and resources, described within the Methods of this paper.

| Resource 1 | Resource 2 (and Higher) | Collaborating on |
|---|---|---|
| NCI-EVS | CDISC | General partnership |
| C-Path | NORD | Jointly working on RDCA-DAP |
| CDISC | NORD | Rare disease TAUG |
| C-Path | CDISC, OMOP | C-Path using CDISC and OMOP standards for its datasets |
| CDISC | OMOP, FHIR | Guidelines on conversions between CDISC, OMOP and FHIR |
| EJP RD | OMOP, ERNs | Mapping between OMOP and CDEs (introduced by ERNs) |
| EJP RD | C-Path, ERNs | Aligning ontologies that are being developed by the ERNs |
| ERNs | EJP RD | FAIRification of data |
| ErkNet (an ERN) | Orphacodes, HPO | ErkNet using Orphacodes for diagnoses and HPO for phenotypes |
| MetabERN, ErkNet (ERNs) | U-IMD | ErkNet and MetabERN contributing to U-IMD standards |
| Phenopackets | OMOP, FHIR | OMOP and FHIR implementations for phenopackets |
| Pharmaceutical companies | CDISC | Regular correspondence when companies encounter non-CDISC data in their studies |

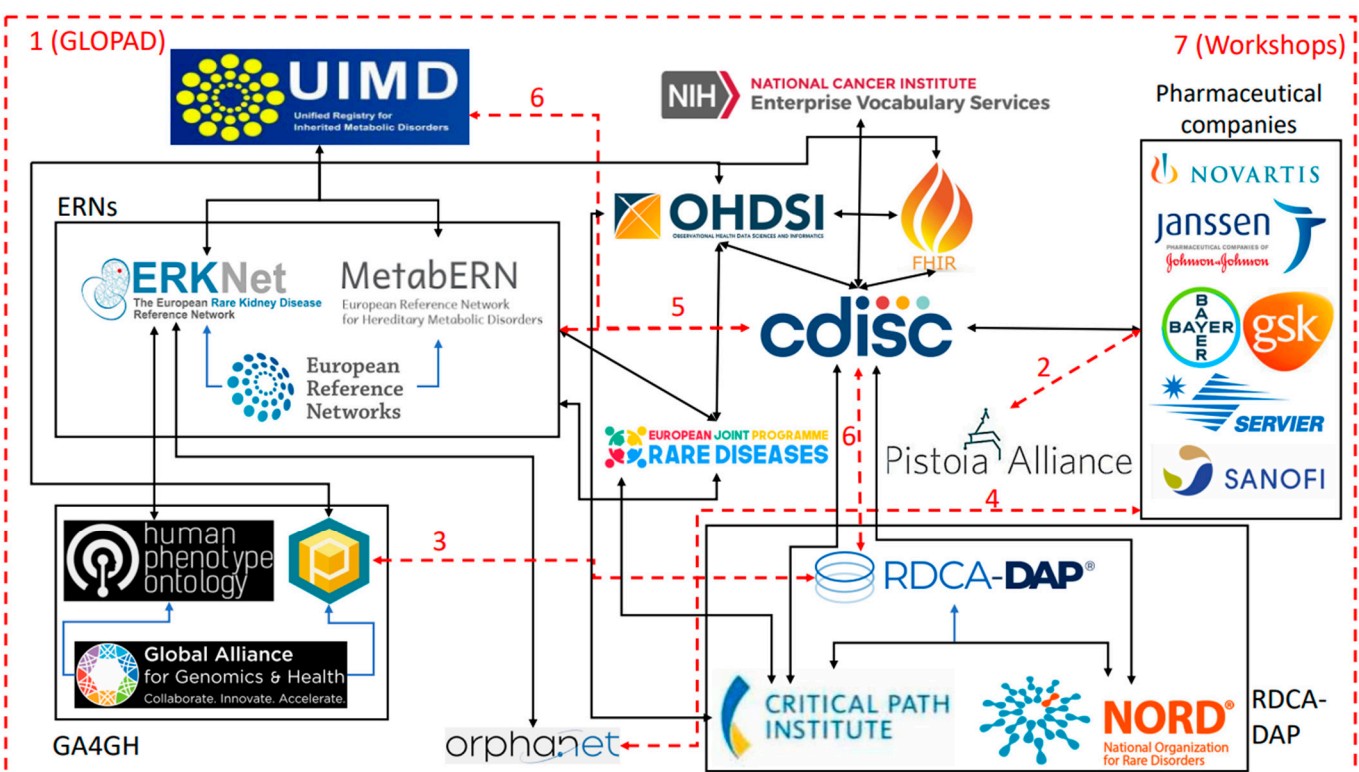

**Figure 3.** Schematic representation of the ongoing and proposed collaborations between large consortia, data resources and data standards and dictionaries. Ongoing collaborations are marked with black double-headed arrows. Single-headed blue arrows denote a smaller resource as part of a larger resource, which are together encapsulated in a box. The box labelled 'Pharmaceutical companies' stands for both industry data dictionaries as well as industries as a collective entity that conducts clinical trials. Red dashed arrows denote the collaborations proposed in the action points with the action number over the arrows. Action points 1 and 7 are shown as a red dashed box around the figure.

*Plan of Action*

A plan of action was developed to address the standardisation and harmonisation of paediatric clinical trial data, with a particular focus on disease-specific data. The plan took forward the collaboration theme, and each action involved two or more of the presented resources. It included seven main action points. Besides the first action, all other actions would be simultaneous.

1. Formalisation of a multi-stakeholder, multi-project user group consisting of members with a wide range of expertise. This group would be responsible for ensuring the progress of all other action points.
2. Use the FAIR4Clin guide for the FAIRification of metadata for industrial and academic paediatric clinical trials.
3. Conduct a Phenopackets pilot that would standardise data from multiple studies in RDCA-DAP and test how pooled data could be used.
4. Using Orphacodes in case report forms for industrial and academic clinical trials.
5. Introducing and educating ERNs with the use of CDISC standards.
6. Exploring applications of the CDISC PUG (potentially using data from all mentioned data sources—RDCA-DAP and U-IMD).
7. Organisation of workshops and educational materials to foster collaboration.

A schematic presentation of existing collaborations and those proposed in the plan of action is shown in Figure 3. It must be noted that these collaborations are not exhaustive, as there may be additional collaborations that become relevant to consider.

## 4. Discussion

As mentioned above, collaboration between initiatives emerged as the major theme during this study. While there are already plenty of existing collaborations between the initiatives and resources, a seven-point action plan was developed, which took collaborations further. In the 20 months since the study was initiated, substantial progress has already been made on the action items. The current status of the seven action points is summarised in Table 3. Detailed discussions follow.

**Table 3.** Summary of the current status of the seven action points. Besides the first action, all other actions would be simultaneous. Green items are close to completion. Yellow items are ongoing. The red item is on hold.

| | Action Item | Current Status |
|---|---|---|
| 1 | Establishment of multi-stakeholder multi-project user group | • GLOPAD established and meeting quarterly.<br>• 2024 workshop in planning. |
| 2 | Use the FAIR4Clin guide for the FAIRification of metadata | • Metadata mapping with FAIR Plus.<br>• Review of FAIR4Clin guide by c4c. |
| 3 | Phenopackets pilot | • Data obtained from RDCA-DAP but paused currently. |
| 4 | Orphacodes in CRFs | • Orphacodes have been recommended in the CCPDD and examples have been provided. |
| 5 | Educating ERNs about CDISC standards | • MetabERN is conducting desk research on the benefits and completeness of CDISC standards for paediatrics.<br>• Course under development on implementing the PUG. |
| 6 | Applications of the Paediatric Use Guide | • Publicising the PUG.<br>• Fortification of CCPDD v2.<br>• Course under development on implementing the PUG. |
| 7 | Workshops and educational material | • Webinar hosted with Pistoia Alliance about FAIR.<br>• Workshop to educate ERNs on data harmonisation approaches in planning.<br>• Course under development on implementing the PUG. |

### 4.1. Action 1—Establishment of the Global Paediatric Data (GLOPAD) Forum

The first action led to the formation of the Global Paediatric Data (GLOPAD) Forum. This group consisted of 56 members at first count, including paediatricians, data standards experts, data scientists, large initiative leads and members, project managers, and experts from industry and academia. The group has been meeting quarterly to take other action items forward. There will be an in-person GLOPAD workshop organised in February 2024.

### 4.2. Action 2—FAIRification of Metadata

The FAIRification of metadata is being carried out in collaboration with the aforementioned FAIRplus project (funding from the Innovative Medicines Initiative 2 Joint Undertaking under grant agreement No. 802750). A metadata profile was created by mapping common metadata elements from 17 existing clinical trial registries/repositories. An initial list of 36 metadata items was reduced to 27 after review by c4c experts. The process and results from this collaboration are documented as a recipe in the FAIR Cookbook [73]. In addition to this, c4c representatives provided reviews of the FAIR4Clin guide and ensured there were references to paediatric data.

### 4.3. Action 3—Phenopackets Pilot

The Phenopackets pilot was started in mid-2022, and data access was obtained for several studies in RDCA-DAP. The goal was to covert data from different sources into a

common format and use the pooled data for some meaningful analysis. While the work had to be paused due to unforeseen circumstances, an academic collaboration is being planned to take it forward. C-Path also plans to initiate work with Phenopackets in 2024.

### 4.4. Action 4—Orpahcodes in CRFs

This action is in its initial stages. References to Orphacodes have been added to the CCPDD and will be available at any trial using terminology from the CCPDD. The CCPDD references CDISC standards, hence creating a link between the two. Adding Orphacodes to CRFs would allow accurate identification of rare disease patients. In the absence of ICD10 codes, patients with a particular disease are identified by a disease name, which can have multiple synonyms—particularly at an international level. This can lead to missing certain patients.

### 4.5. Action 5—Educating ERNs about CDISC Standards

Introducing CDISC standards to the ERNs is a first step towards getting academic institutions acquainted with CDISC standards. MetabERN is conducting desk research among ERNs about the benefits and completeness of CDISC standards in paediatrics. One task has involved translating CRFs to CDISC standards.

### 4.6. Action 6—Applications of the PUG

The PUG was heavily promoted within the c4c consortium, including all its 20 national hubs and over 150 sites. It was also presented to CDISC users at the European CDISC Interchange in April 2023. The first application of the PUG was used to fortify the next version of the CCPDD. It added forty new data items to the CCPDD, and ten new categories were created. The aforementioned course will add to the understanding and utilisation of the PUG.

### 4.7. Action 7—Workshops and Educational Materials

A tentatively planned workshop will evaluate the potential of using ERN registries for paediatric clinical research in the 24 therapeutic areas they cover. The focus will be on additional data items the registries would need to collect to support trial feasibility assessment, paediatric implementation plans, and the potential usage of the registry as a control arm. c4c developed a webinar with the Pistoia Alliance to highlight the importance of taking a FAIR approach to paediatric data [74]. The PUG implementation course is a part of this action as well.

#### 4.7.1. Wider Dissemination

Despite the progress, most of the actions have been implemented within c4c and its collaborators. For a higher global impact, the dissemination needs to extend to the wider community. This paper is the first step in that direction. The action items in this paper are not restricted to the mentioned resources and are applicable to a variety of resources. For example, data from RDCA-DAP can be replaced by data from YODA for a pilot study. The PUG is available publicly and can be used for any application. Going forward, the GLOPAD will serve as the main tool for wider dissemination. Pending approval, the GLOPAD will be publicised on the c4c website. Anyone can apply to join, and further information will be provided on the website.

#### 4.7.2. Future beyond c4c

While the funding period for c4c will continue until April 30, 2025, the data standardisation and harmonisation strategies will remain relevant. A new legal entity—conect4children Stichting—a non-profit organisation aiming to sustain the work achieved by c4c, has already been established and is based in Utrecht, Netherlands. A stichting is a limited liability legal entity in the Netherlands with no owner (or shareholders) and governed by a board of directors. The GLOPAD is expected to continue its operations under the c4c Stichting,

with actions 2, 4, 5, and 6 being transferred to the stichting. New actions may be added when necessary. Some long-term actions that were eventually not included in the plan of action can likely be addressed in the future.

### 4.7.3. Challenges

While a path was identified for the standardisation of disease-specific data, major challenges remain. The collaborative strategy assumes that informed consent has been given by patients; however, informed consent wordings can prevent patient-level data from being shared openly—particularly across international borders [75]. Even after informed consent is obtained, institutions want to protect their own intellectual property and often hesitate or delay sharing of data due to scientific or business competition [76]. Once the data is shared, non-standard data formats can be a challenge. Several institutions use proprietary formats that hold little value outside of the institution. This is particularly true when pharmaceutical companies need to create internal dictionaries due to the lack of paediatric-specific terminologies in CDISC. While all companies are committed to interoperability, the creation of proprietary internal dictionaries leads to a lack of standardisation as implementation guidelines can be interpreted differently by individual companies. Moreover, substantial data currently under consideration for sharing by institutions is legacy data, and significant costs are associated with sharing these data sets, even if it is permissible. Costs include digitalisation, anonymisation, format conversion, curation, etc. Institutions may be unwilling to undertake these costs, particularly in the absence of short-term benefits. Finally, there is a lack of knowledge about existing initiatives for data sharing. This prevents the sharing of data even when they are available, undermining considerable global efforts to improve the effectiveness and efficiencies of paediatric clinical trials.

Despite the discussed resources being far-reaching, there are several underrepresented diseases among them—particularly individual rare diseases. This is largely due to a lack of guidelines regarding their inclusion in registries, a lack of initiatives tackling individual diseases, and a lack of clinical expertise in diagnosis, management, and outcome measures. ERNs were a first step towards enabling clinicians to access rare disease knowledge generated throughout Europe; wider efforts are still in nascent stages.

### 4.7.4. Limitations

There are obvious limitations in the review and action plans presented here. Since the review mainly focussed on the resources and initiatives of c4c connections, it is not exhaustive. For example, the Data Analysis and Real-World Interrogation Network (DARWIN) project delivers real-world evidence across diseases and populations in addition to supporting regulatory decision-making. The plan of action did not involve other important stakeholders—patients (along with their caregivers) and policy-makers. For any data to be interoperable, they must be shared beyond the institution it was acquired in. Patients need to be comfortable with the process. This is particularly true for rare disease where just the diagnosis can be identifying information, making anonymisation difficult. For actions that are far-reaching and have global significance, policy-makers need to be involved in the planning. Policy makers include public health agencies who can support economic assessments for decisions related to healthcare.

## 5. Conclusions

We identified the need for standardised disease-specific data and reviewed multiple initiatives, data resources, and data standards that could be utilised for this purpose. We describe a plan of action that can act as a steppingstone to the long-term standardisation goals.

**Author Contributions:** Conceptualisation, A.S., V.H., S.N., F.B., R.L. and V.S.; Formal analysis, A.S. and E.D.; Funding acquisition, V.S.; Investigation, A.S., V.H. and R.L.; Methodology, A.S., V.H., E.D., S.H., R.C., R.W., J.O., P.N.R., E.G.N., T.L., G.N., N.M., S.W., A.P., R.G., M.W., T.B., J.L., L.P., K.B., K.A.H., S.S.S., J.O.B.J., G.R., M.A.K., C.P., D.B., S.N., F.B., R.L. and V.S.; Project administration, V.H., A.P., J.T. and R.L.; Resources, S.H., R.C., R.W., J.O., P.N.R., E.G.N., T.L., G.N., N.M., K.B., J.O.B.J.,



M.A.K., C.P., D.B., F.B. and R.L.; Supervision, V.H., D.B., S.N., F.B., R.L. and V.S.; Validation, A.S., V.H., E.D., S.H., R.C., R.W., J.O., P.N.R., E.G.N., T.L., G.N., N.M., S.W., A.P., R.G., M.W., T.B., J.L., L.P., K.B., K.A.H., S.S.S., J.O.B.J., G.R., M.A.K., J.T., C.P., D.B., S.N., F.B., R.L. and V.S.; Visualisation, A.S., E.D. and S.H.; Writing—original draft, A.S., V.H., E.D., S.H., R.C., R.W., J.O., P.N.R., T.L., G.N., N.M., A.P., J.T., D.B., S.N., F.B., R.L. and V.S.; Writing—review and editing, A.S., V.H., E.D., S.H., R.C., R.W., J.O., P.N.R., E.G.N., T.L., G.N., N.M., S.W., A.P., R.G., M.W., T.B., J.L., L.P., K.B., K.A.H., S.S.S., J.O.B.J., G.R., M.A.K., J.T., C.P., D.B., S.N., F.B., R.L. and V.S. All authors have read and agreed to the published version of the manuscript.

**Funding:** This project has received funding from the Innovative Medicines Initiative 2 Joint Undertaking under grant agreement no. 777389. The Joint Undertaking receives support from the European Union's Horizon 2020 research and innovation programme and EFPIA.

**Institutional Review Board Statement:** Not applicable.

**Informed Consent Statement:** Not applicable.

**Data Availability Statement:** No new data were created or analyzed in this study. Data sharing is not applicable to this article.

**Acknowledgments:** The authors would like to thank Ian Harrow and the c4c project leadership team for reviewing the paper.

**Conflicts of Interest:** The authors declare no conflicts of interest.

**Disclaimer:** The publication reflects the authors' view and neither IMI nor the European Union, EFPIA, or any Associated Partners are responsible for any use that may be made of the information contained therein.

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
