# Peer review of "Learning from conect4children: A Collaborative Approach towards Standardisation of Disease-Specific Paediatric Research Data"

_data, 2024_

Round 1

Reviewer 1 Report (Previous Reviewer 3)

Comments and Suggestions for Authors

As in the previous review this article is very interesting, the authors have reduced some how the abstract and in my opinion can be published as is.

Author Response

Reviewer 1:

As in the previous review this article is very interesting, the authors have reduced some how the abstract and in my opinion can be published as is.

Response: Thank you for the encouraging words.

Reviewer 2 Report (New Reviewer)

Comments and Suggestions for Authors

The authors present a very interesting approach in harmonizing and standardizing patient data and records in depicting and optimizing clinical outcomes in the pediatric population.

However, it is unclear whether the authors have included public repositories of previously published data regarding clinical studies involving pediatric participants, as well as preprints and other available data repositories. THe authors are advised to elaborate more on this.

Additionally, I suggest the authors to extend the Limitations section to other issues such as over- or underrepresentation of certain conditions due to lack of guidelines for certain management plans, especially in rare diseases, and possible strategies to mitigate this. I believe clinical expertise in assessing the relevance of data collected in terms of clinical significance (outcomes) should be more emphasized.

Author Response

Reviewer 2:

Comment: The authors present a very interesting approach in harmonizing and standardizing patient data and records in depicting and optimizing clinical outcomes in the pediatric population.

Response: Thank you for the encouraging words. 

Comment: However, it is unclear whether the authors have included public repositories of previously published data regarding clinical studies involving pediatric participants, as well as preprints and other available data repositories. THe authors are advised to elaborate more on this.

Response: This is an important point. In fact, we have a parallel study analysing repositories with paediatric provisions. The manuscript is currently under review (second round) in the same journal (Data). A pre-print of the manuscript prior to the first round of revisions can be viewed here: https://drive.google.com/file/d/195z8KM27Xldxa198ftQesbi6-IQqfMqx/view?usp=sharing (link is shared only for review purposes and will be removed once a decision on this paper has been made). The text of has been modified (lines 270 to 279) to make this clear. Depending on if/when the above paper gets accepted, we may be able to cite it in this paper.

Modified text:  The UK National Neonatal Research Database contains a standard data extract from the Electronic Patient Records of all admissions to National Health Service neonatal units from 2007 to the present. Data are quality assured and curated prior to entry into the database and are mapped to the OMOP Common Data Model (see below). Access processes are through the Health Data Research UK Alliance Gateway [49]. This is a rare case of a strictly paediatric repository. Hence, repositories that primarily contain adult patient data but have provisions for paediatric data are of specific interest. The review and analysis of such repositories is being carried in a parallel c4c study. While going over every analyzed repository is beyond the scope of this paper, a few examples are discussed below.

Comment: Additionally, I suggest the authors to extend the Limitations section to other issues such as over- or underrepresentation of certain conditions due to lack of guidelines for certain management plans, especially in rare diseases, and possible strategies to mitigate this. I believe clinical expertise in assessing the relevance of data collected in terms of clinical significance (outcomes) should be more emphasized.

Response: We have added a paragraph addressing this and highlighted the role of ERNs in sharing rare disease knowledge across Europe. However, we placed it within the ‘Challenges’ subsection (lines 573-578) as ‘Limitations’ was primarily intended for the limitations of the paper rather than the field of study.

Modified text: Despite the discussed resources being far-reaching, there are several underrepresent-ed diseases among them – particularly individual rare disease. This is largely due lack of guidelines regarding their inclusion in registries, lack of initiatives tackling individual diseases and lack of clinical expertise in diagnosis, management, and outcome measures. ERNs were a first step towards enabling clinicians accessing rare disease knowledge generated throughout Europe, wider efforts are still in nascent stages.

This manuscript is a resubmission of an earlier submission. The following is a list of the peer review reports and author responses from that submission.

Round 1

Reviewer 1 Report

Comments and Suggestions for Authors

The need for standardisation of paediatric data is well made. However, the paper often appears to be a report of work with details of the workshop sessions (see section 2.1). Section 3.1Plan of action is OK for a report of work to the funders but inappropriate for a scientific paper.

Section 2.2 would be better as a table listing the large collaborations.

The authors are also too self-praising,eg "c4c has been at the forefront in developing resources to enable the pooling of data"(line 92). This is upto others to decide.

The authors need to dramatically shorten the paper - keeping essential information only. This would make the paper more useful and more likely to be read.

Author Response

We thank the reviewer for their comments. The paper has improved as a result of changes made from addressing them. Here we respond to each point and explain where changes were made or reasons for not implementing them.

Reviewer 1

The need for standardisation of paediatric data is well made. However, the paper often appears to be a report of work with details of the workshop sessions (see section 2.1). Section 3.1Plan of action is OK for a report of work to the funders but inappropriate for a scientific paper.

Response: This was an important point that we debated while working on the paper. We do agree this isn’t a typical research paper. We have mentioned in our cover letter that we would be happy to consider this as a review or position paper. We considered not mentioning the workshop at all but decided to include it as that was what brought all these resources together and help us understand the deep collaboration network that existed within. As mentioned in Reviewer 2’s comments, the plan of action is a part of wider dissemination, and we would hope that others would be encouraged to take up similar actions with different resources. This is now mentioned explicitly in the Discussion.

Section 2.2 would be better as a table listing the large collaborations.

Response: A table has been added to highlight the main resources of the large initiatives presented in this paper. The text has been retained alongside (please also see response to the comment on the length).

The authors are also too self-praising,eg "c4c has been at the forefront in developing resources to enable the pooling of data"(line 92). This is upto others to decide.

Response: Apologies for the text coming across this way. We have made appropriate changes, including the sentence mentioned.

The authors need to dramatically shorten the paper - keeping essential information only. This would make the paper more useful and more likely to be read.

Response: While we agree the paper is on the longer side, we did want to take a deeper dive into the mentioned resources. For example, just describing the EJP-RD would be repeating the main information available on its website. We have gone into two resources of EJP-RD – the single access virtual platform and the Innovation Management Toolbox, that may be relevant to disease-specific standardization. The journal was selected keeping the higher wordcount in mind. ‘Data’ recommends a wordcount of 3000 – 12000, while this revision is about 7200 words. The length of the paper is also seems artificially inflated due to the 80+ references.

Reviewer 2 Report

Comments and Suggestions for Authors

The article is a very interesting approach towards a very important problem of antibiotics and therapies that exist already and also that are in development for children. The approach of the authors is to create a network that would better connect bench to bedside and also to connect different approaches. It is important to start developing such an approach and also to include all existing resources for the good of children.

It would be however interesting to the readers how the authors will disseminate their work in the community and how will these network support all the developments of the proposed research. The novelty of the idea for a complex research group is an very interesting approach and may be considered at the main strong point while the dissemination and also the difficulties that may be encountered represent a weak point.

Author Response

We thank the reviewer for their comments. We have addressed the mentioned point and we believe the paper has improved as a result.

Reviewer 2

The article is a very interesting approach towards a very important problem of antibiotics and therapies that exist already and also that are in development for children. The approach of the authors is to create a network that would better connect bench to bedside and also to connect different approaches. It is important to start developing such an approach and also to include all existing resources for the good of children.

It would be however interesting to the readers how the authors will disseminate their work in the community and how will these network support all the developments of the proposed research. The novelty of the idea for a complex research group is an very interesting approach and may be considered at the main strong point while the dissemination and also the difficulties that may be encountered represent a weak point.

Response: This is an important point and thank you for bringing it to our attention. Dissemination was one of the primary objectives of the paper itself though it was not explicitly mentioned. We have now added a small paragraph in the Discussion talking about wider dissemination. We do want to point out that c4c does not have the authority to mandate how clinical trial data are collected, so dissemination is limited to raising awareness and providing possible solutions (in form of action points).

Reviewer 3 Report

Comments and Suggestions for Authors

An interesting manuscript describing outcomes from the conect4children (c4c) project. An EU funded project under the umbrella of Horizon 2020.

This project was established to facilitate the development of new drugs and other therapies for paediatric patients. This manuscript can serve as a complete guide for all efforts towards this aim.

The manuscript is well written and appropriate for the journal.

Only one comment that may help to improve this work:

1.       The abstract is too long, propose to reduce at least by 50% and focus on c4c main outcomes.

Author Response

We thank the reviewer for their comments. We have addressed the mentioned point and we believe the paper has improved as a result.

Reviewer 3

An interesting manuscript describing outcomes from the conect4children (c4c) project. An EU funded project under the umbrella of Horizon 2020.

This project was established to facilitate the development of new drugs and other therapies for paediatric patients. This manuscript can serve as a complete guide for all efforts towards this aim.

The manuscript is well written and appropriate for the journal.

Only one comment that may help to improve this work:

  1. The abstract is too long, propose to reduce at least by 50% and focus on c4c main outcomes.

Response: The abstract has been substantially reduced in length.

Round 2

Reviewer 1 Report

Comments and Suggestions for Authors

The paper still reads like a report of work and remains too long.

The authors have added a table, but have not amended much of the text, apart from the abstract.

The paper needs a major rewrite- sorry

Author Response

The paper has been reduced by 20% in length (~1500 words: from 7254 to 5770). This is now less than half the word limit specified by the journal (12000 words).

To address the paper reading like a report, all mentions of the workshop have been removed and the resources are presented as tools for addressing disease-specific standardisation. The action items have been retained in the results but all discussion regarding them have been moved to the Discussion. While we considered removing the action items altogether, without them the paper would just state a problem, suggest possible resources but present no solutions. We hope the actions presented will raise awareness about possible solutions and inspire larger organizations to take this forward (mentioned in the ‘Wider Dissemination’ subsection).